# A Multi-Sensor System for Sea Water Iodide Monitoring and Seafood Quality Assurance: Proof-of-Concept Study

**DOI:** 10.3390/s21134464

**Published:** 2021-06-29

**Authors:** Alessandro Zompanti, Simone Grasso, Anna Sabatini, Luca Vollero, Giorgio Pennazza, Marco Santonico

**Affiliations:** 1Unit of Electronics for Sensor Systems, Department of Engineering, Campus Bio-Medico University of Rome, 00128 Rome, Italy; a.zompanti@unicampus.it (A.Z.); a.sabatini@unicampus.it (A.S.); 2Unit of Electronics for Sensor Systems, Department of Science and Technology for Humans and the Environment, Campus Bio-Medico University of Rome, 00128 Rome, Italy; s.grasso@unicampus.it (S.G.); m.santonico@unicampus.it (M.S.); 3Unit of Computational Systems and Bioinformatics, Department of Engineering, Campus Bio-Medico University of Rome, 00128 Rome, Italy; l.vollero@unicampus.it

**Keywords:** iodine, iodide, electrochemical sensors, gas sensors, seafood quality, sea water

## Abstract

Iodine is a trace chemical element fundamental for a healthy human organism. Iodine deficiency affects about 2 billion people worldwide causing from mild to severe neurological impairment, especially in children. Nevertheless, an adequate nutritional intake is considered the best approach to prevent such disorders. Iodine is present in seawater and seafood, and its common forms in the diet are iodide and iodate; most iodide in seawater is caused by the biological reduction of the thermodynamically stable iodate species. On this basis, a multisensor instrument which is able to perform a multidimensional assessment, evaluating iodide content in seawater and seafood (via an electrochemical sensor) and discriminating when the seafood is fresh or defrosted quality (via a Quartz Micro balance (QMB)-based volatile and gas sensor), is strategic for seafood quality assurance. Moreover, an electronic interface has been opportunely designed and simulated for a low-power portable release of the device, which should be able to identify seafood over or under an iodide threshold previously selected. The electrochemical sensor has been successfully calibrated in the range 10–640 μg/L, obtaining a root mean square error in cross validation (RMSECV) of only 1.6 μg/L. Fresh and defrosted samples of cod, sea bream and blue whiting fish have been correctly discriminated. This proof-of-concept work has demonstrated the feasibility of the proposed application which must be replicated in a real scenario.

## 1. Introduction

Iodine is a trace chemical element fundamental for a healthy human organism [1]. Iodine is present in seawater and seafood [2], and its common forms in the diet are iodide and iodate [3]; most iodide in seawater is caused by the biological reduction of the thermodynamically stable iodate species [4]. Thus, iodide could represent both an index of sea water and environmental conditions (also because its cycle impacts on air quality [5]) and it is also important for human health [6]. Iodine deficiency affects about 2 billion people worldwide causing from mild to severe neurological impairment, especially in children [7,8,9]. Nevertheless, an adequate nutritional intake is considered the best approach to prevent such disorders. Although iodine content in eggs, fish, meat, dairy and grain products would be enough to satisfy the recommended daily intake (100–300 μg) [10], the lack of an adequate diet remains the major cause of its deficiency, especially in developing countries. Iodine is an essential component of the thyroid hormones thyroxine (T4) and triiodothyronine (T3), which regulate many important biochemical reactions and are critical determinants of metabolic activity. Particularly, during pregnancy and early infancy, thyroid hormones deficiency can cause irreversible effects including neurodevelopmental deficits and growth retardation [11]. Nevertheless, iodine deficiency disorders (IDD) are among the easiest and least expensive of all nutrient disorders to prevent. Therefore, a programme based on salt iodization for people in daily consumption has been considered an adequate solution [12]. Like any other health interventions, monitoring of salt iodization programmes is essential to ensure that it is functioning as planned and to provide information needed to take corrective actions if necessary. For this reason, screening campaigns should be taken at both the production level, reflecting the industrial quality assurance of procedures, and the consumer level, which represents the iodine status intake in the population. Median urinary iodine is the main indicator to be used to assess iodine status of a population as the human organism excretes more than 90% of dietary iodine in the urine [11]. Many analytical techniques exist, varying from very precise measurement with highly sophisticated instruments, to semi-quantitative low-tech analysis; however, almost all methods depend on digestion or other purification steps that require high-end laboratory equipment and expert personnel [13]. Furthermore, to provide a reasonable estimation of the iodine status, a population monitoring program should be high-frequency and based on a wide pool of samples. The strategy here proposed has a double point of view as starting point: sea water and seafood, with the aim of supporting consumer’s conscious choice and, possibly, to support an environment–food–humans monitoring campaign. Thus, a multisensor instrument which is able to perform a multidimensional assessment is needed. To this scope, in this work, a multi-focus and multidimensional approach is proposed: the first focus is on seafood, which represents an optimal dietary option for iodide consumption [14]; the second focus is on sea environment (seafood supplying source) which is naturally rich in iodide concentration. On this basis, a multisensor instrument [15], which is able to evaluate iodide content in seawater and seafood (via an electrochemical sensor) and to discriminate when the seafood is fresh or defrosted (via a Quartz Micro balance (QMB)-based volatile and gas sensor), is strategical for seafood quality assurance. Moreover, an electronic interface has been opportunely designed and simulated for a low-power portable release of the device, which should be able to identify seafood over or under an iodide threshold previously selected. The electrochemical sensor has been successfully calibrated in the range 10–640 μg/L, obtaining a root mean square error in cross validation (RMSECV) of only 1.6 μg/L. Fresh and defrosted samples of cod, sea bream and blue whiting fish have been correctly discriminated. The novelty of the method proposed is given by the multidimensional approach (many features), by the multisensory application (QMBs and cyclic voltammetry), and by the step-by-step monitoring strategy (from the sea to the consumer). Moreover, the voltammetric liquid sensor designed and calibrated for iodide, compared to more precise and complex methods [16,17,18,19], has to be simpler, portable and cost effective. Regarding seafood evaluation based on volatiles, many papers are available in the literature [20,21,22,23,24]. In general, different working principles have been utilized and many instruments have been developed and tested [21,22,23,24]. In particular, a multidimensional approach similar to the one here proposed has been experimented in [20]. Moreover, this experiment was focused on fish quality but not specifically addressed to health-state objectives such as that regarding iodide consumption.

## 2. Materials and Methods

In Figure 1 a general overview of the experimental set-up is shown. This is a schematic representation of a real scenario application for quality assurance of seafood and for the sea environment origin. In this work, three steps of this flow chart have been implemented. Then, each subsection of this paragraph reports the details of the methods used for each of the three tests.

Seafood should undergo a step by step procedure in which, first, an instrument discriminates if the product is fresh or defrosted; whenever the seafood was fresh, its iodide content is compared against a selected threshold (depending on the medical indication given for the target population of consumers); the entire process should provide a certified label claiming the seafood quality assessment. The steps 1, 2 and 3 have been developed in the experiment reported in this paper.

### 2.1. Iodide Calibration

The instrument used for calibration of Iodide is the BIONOTE-L (BIONOTE for Liquid) [15]; it is a voltammetric sensor composed of a disposable screen-printed electrode (SPE DRP-250BT, Metrohm, Herisau, Switzerland) probe (Working: Gold; Counter: Platinum; Reference: Silver), and of a dedicated electronic interface devoted to supply a triangular waveform from +1 V to −1 V as input, and to record the current generated by the induced oxi-reduction phenomena of the analytes dissolved in the aqueous media. The current is then converted in voltage by a trans-impedance circuit and saved as output data. The frequency of the input signal was set to 0.01 Hz and 500 output values were collected for each measuring cycle.

The calibration of the voltammetric sensor against iodine was performed using potassium iodide (221945, Sigma Aldrich, St. Louis, MO, USA) chemical standard dissolved in bi-distilled water at the desired concentration. Each sample solution was fresh prepared just before being analysed to avoid any potential oxidation effect. Eight concentration values of iodide were tested: 0, 20, 40, 80, 160, 320, 640 μg/L. Three measurements were performed for each concentration level. Standard deviation in terms of percentage with respect to the response output range from 0.1% to 2%, which was good reproducibility.

Partial least square (PLS) analysis coupled with the leave-one-out criterion as a cross-validation method were employed to obtain all the predictive models onto the calibration data. All the predictive models were calculated using PLS-Toolbox (Eigenvector Research Inc., Manson, WA, USA) in the Matlab Environment (The MathWorks, Natick, MA, USA).

### 2.2. Seafood Analysis

Three species of fish were the seafood selected for this experiment: cod, sea bream and blue whiting fish. Five samples were collected for each fish species. Each measurement was performed in duplicate. A fillet of 5 g was extracted from each sample. Each fillet was placed in a Petri plate (20 mm of diameter). The volatiles in the headspace over the fillet were collected via a dedicated cylinder and conveyed into the measure chamber of the gas sensor array used in this work (see Figure 2), at a flow of 0.4 L/min. Nitrogen was the carrier gas. The gas sensor array used was the BIONOTE-V (BIONOTE for Volatiles) [15]. It is an array of seven quartz microbalances functionalized with seven different anthocyanins. The sensors are seven slices of AT-cut quartzes oscillating at a resonance frequency of 20 MHz. Each quartz is driven by a dedicated oscillator circuit. The sensing materials covering the crystals’ surface are seven anthocyanins extracted from different plants, fruits and flowers (see ref. [15] for the details of the functionalization). Fish fillets were firstly extracted and measured when the fish was fresh (freshly caught). Then all the samples of fishes were frosted (following the regulation stated in EU 1276/2011). Three days later the fishes were defrosted, fillets were extracted (see the procedure above described) and analyzed. An analysis of variance (ANOVA) test was performed on the BIONITE-V outputs to assess if defrosted and fresh samples are different with statistical significance (*p* < 0.05). Considering that each measurement was composed of an array of seven numbers and that the ANOVA test was a mono-dimensional test, data were treated to synthesize the olfactive intensity of each measure in a single number. This number was calculated as the sum of all the frequency shifts registered by each sensor for the same sample. The approach for the analysis of fish volatiles was the multidimensional measurement based on an array of sensors: the BIONOTE-V, composed of seven QMBs, functionalized with non-selective sensing materials. This approach is typical of the electronic nose paradigm [25,26,27]. On this basis, odour has multidimensional information which is well represented by a fingerprint. Considering that the main objective was the discrimination among two conditions (fresh/defrosted) the multidimensionality of the output data was abruptly reduced to one dimension in order to perform the most simple statistical test to achieve the discrimination goal (the simpler the data analysis technique, more reliable is the result). This index was defined as olfactive intensity, being the sum of the response given by all the Volatile Organic Compounds (VOCs) present on the headspace of the samples analyzed. Moreover, the multidimensional fingerprint is the best way to represent odour-like data, thus also principal component analysis and partial least square discriminant analysis were performed on the BIONOTE-V data.

### 2.3. Threshold System

The calibration test indicated that iodide can be detected and quantified (see the results section). Two voltage values could be defined by the observation of the voltammograms: −0.87 V and −0.5 V. For them, an increase in the current peaks was registered, directly proportional to the iodide concentration increase (see Section 3.1).

In order to make the detection process rapid and easy to perform, a portable and low power configuration of the device is essential. This is feasible when the detection system is designed just for a specific compound (iodide in this case) and it is addressed to a specific threshold monitoring (meaning specific concentration levels). The schematic of the equivalent electronic circuit is reported in Figure 3.

This circuit is powered by a fixed voltage (V1), corresponding to the input voltage selected via voltammetric calibration, and which can be granted by a button cell battery but which can be also realized with a piezoelectric device, requesting to provide a voltage supply only during the measuring phase. A detailed description of the circuit and the results obtained with the simulation performed at the voltage values defined above, are reported in the results section (Section 3.2). The simulation was performed with the software Multisim 12 (National Instrument, Austin, TX, USA).

## 3. Results

The results of the sensor in iodide calibration are presented here as the first step, because it is mandatory for the realization of the threshold system, presented as conclusive sub-section of the results. The second sub-section reports the results obtained on the discrimination of fresh and defrosted fish: without the system’s ability in fish freshness discrimination, seafood analysis in terms of iodide is necessary but not sufficient to achieve the goal of the work.

### 3.1. Iodide Calibration

The average of the last measuring cycles across the experimental triplicates was calculated and the output signals of all samples was compared (Figure 4). The plot of the cyclic voltammograms clearly shows an increment of the signal across specific regions of the input scan, suggesting a correlation with the iodine concentration.

Observing Figure 4 there are two voltage values in particular for which the current peak increases with the increase of iodide concentration level. These voltage values are −0.87 V and −0.5 V.

Due to the multivariate nature of the acquired data, a PLS predictive model has been calculated on the overall dataset. As can be seen in Figure 5, the BIONOTE-V was able to discriminate iodine concentration in simple solutions with a RMSECV of only 1.6 μg/L.

Despite this is a preliminary study and further in-depth investigations are required, the impressive results obtained aim to propose the BIONOTE-L as an easier, quicker, cheaper and innovative high-throughput alternative to traditional iodine analytical methods.

### 3.2. Threshold System

The electronic circuit can be studied considering a simplified model (Figure 3 in methods section). In order to describe the circuit, three characteristic points can be selected for a deeper analysis. The first point is the reference point placed between the two resistances (R_1_ and R_2_). This is the reference voltage which does not change during the measurement. The resistance R_3_ (series with R_4_) plays the role of determining the current value flowing into R_4_, which represent one of the interface impedance with the electrodes. The voltage given by the series of R_3_ and R_4_ represent a reference value. It is worth remarking that this current cannot flow towards V1 and can be used as a current feedback between the three nodes representing the three-electrodes probe. Thus the interaction of the electrodes with the solution under analysis is mainly represented by the resistances R_1_ and R_2_. It is expected that, if the model of the circuit correctly represents the real situation, when the input voltage is fixed, different current values must be observed for different values of R_1_ and R_2._ The simulation had the goal of verifying the correspondence of the current values with those reported in Figure 4, also assessing if the relative values of R_1_ and R_2_ were comparable with a real practical condition of interest. The experimental results obtained for the iodide detection has suggested that two specific voltage inputs can be used as exciting signal: −0.87 V and −0.5 V. Clearly a fixed voltage produces a specific current value in output. In Table 1 and Table 2 the numerical results of the simulation are reported.

It emerges from Table 1 and Table 2 that the values of resistance selected accounts for different features of the system simplified by the electrical model: R_1_ is relative to the commercial electrode (which is always the same, and which does not depend on the voltage applied and on the salt concentration); R_4_ is constant for each specific voltage reference, because it accounts for the interaction of the voltage excitation with the solution; R_2_ is of the order of magnitude of a solution of distilled water with different concentration of a single salt [28], in fact it decreases with the increase of iodide concentration. It is evident that this electronic configuration is auto consistent, and it does not need of a specific power supply for active elements because, in this simplified configuration, the operational amplifiers are not present. This is possible just because, after a calibration process, a specific input voltage has been identified and a wave form generator is not necessary. Cleary the current can change also if the solution modifies its characteristic, for example modifying the iodine concentration. In this case we can be assume the impedances R_1_ change. The sensitivity of the system is given from:$$\frac{dI}{dR1}=-\frac{\Delta V_{R1}}{R_{1}^{2}}$$

We can observe that the sensitivity increases with a reduction of R_1_ that depends on the concentration of iodine. So we can assert that if the concentration of iodine increases, the relative impedance of the solution decrease because the iodine increase the conductivity of the solution with a reduction of impedance. The two capacitances placed in parallel to R_1_ and to R_2_ are the contact-capacitance always present in a configuration of this type. Furthermore, these capacitances are necessary when we consider the noise effect. It is note that for a resistance theoretically infinite the thermal noise can assume infinite values. This assumption is not realistic, obviously. Estimating the performance of this strategy in order to compare it with the BIONOTE-L, a theoretical approach could be used. The concentration–current relationship appears to be linear and the sensitivity is of 0.18 μg/L. Considering the minimum detectable current as 0.1 μA (calculated considering the Johnson Thermal Noise), the resolution is of 0.55 μg/L, which gives a final LOD of 1.66 μg/L, the same value of the estimated RMSECV in the multidimensional approach.

### 3.3. Seafood Analysis

The approach for the analysis of fish volatiles was the multidimensional measurement based on an array of sensors: the BIONOTE-V, composed of seven QMBs, functionalized with non-selective sensing materials. This approach is typical of the electronic nose paradigm. On this basis, odour is multidimensional information which is well represented by a fingerprint. The fingerprints registered for the fish samples of each species (codfish, sea bream, blue whiting) in the two conditions (fresh and defrosted) are characterized by different shapes (typical of the species) and this typical shape increases (in terms of area) from the defrosted to the fresh sample: see Figure 6.

The main objective was the discrimination among two conditions (fresh/defrosted). To achieve this goal the multidimensionality of the output data was reduced to one dimension in order to perform the most simple statistical test, because the differences among fresh and defrosted samples must be proven by a statistical significance. ANOVA, calculated on the olfactive intensity (defined in the methods section), confirmed the difference among the two conditions with a *p* < 0.01 for all the three species. This significance is evident by the boxplots reported in Figure 7.

This index was defined as olfactive intensity, being the sum of the response given by all the VOCS present on the headspace of the samples analyzed. Besides, the multidimensional fingerprint is the best way to represent odour-like data, also PCA and PLS discriminant analysis were performed.

Considering that three different species of fish have been analyzed and that a significant number of samples have been measured for each species, the first objective of PCA analysis was the ability of the sensors in discriminating the different species.

The score plot of the first two principal components (PCs), reported in Figure 8, puts in evidence three different clusters pertaining to the three fish species.

PC1 reports the 72% of the explained variance, thus the most part of the informative content of the volatile fingerprint of each sample: along the direction addressed by PC1 it is possible to discriminate the three species of fish analyzed well. The aim of this experiment was not fish species identification, but the discrimination among fresh and defrosted fish. Note that demonstrating sensor ability in the discrimination of attributes related to the fish typology makes its ability to recognize fresh fish samples more reliable. Observing Figure 8 is worth noting that the direction from right to left of the PC1 indicates three different species of fish in increasing order of fat content [29]. This is significant in terms of volatile emission, because fat content in a food matrix has the power of retaining the most of the volatile compounds.

Looking at the results given by this PCA model and considering the ability of discriminating fresh and defrosted samples of fish, independently on the fish species, the next step was to ask data whenever fishes of the same species had different volatile fingerprints in case of defrosting actions. In Figure 9 are three score plots relative to the three species (panel a: codfish; panel b: sea bream; panel c: blue whiting).

It is worth noting that, considering that each PCA scores plot reported in Figure 9 is referred to many samples of the same species, the main difference between the samples is the fact that some of them are fresh and some others are defrosted. This information, indeed, is visible along the PC1, which is the most significant PC for each of the models.

Considering that PCA is an explorative method, the evidence observed in Figure 8 and Figure 9 must be confirmed by using a multivariate classification technique.

Partial least square discriminant analysis was applied on the same datasets used for PCA analysis. Two models were built. The first model aimed at discriminating fish species. The second model aimed at discriminating fresh and defrosted samples of fish, independent of the species. Both the models gave a 100% rate of correct classification.

## 4. Discussion

A three-step process was followed in this work. The research question was the possibility of setting up a method for the application of a multi-sensor system for sea water iodide monitoring and seafood quality assurance. The multi-sensor system was composed of two devices: one for volatiles analysis and one for liquid analysis. The first (BIONOTE-V) demonstrated the ability to discriminate fresh and defrosted seafood (limited to the case study here proposed). The second (BIONOTE-L) was correctly and effectively calibrated to the concentrations of interest of iodide. Thus, they can be used together for seafood quality assurance, also monitoring iodide content. Moreover, a simplified version of the voltammetric liquid sensor could support its utilization on field. A circuit topology was designed and simulated for this ‘portable’ device. The simulation confirmed the effectiveness of the circuit strategy. This approach could be very useful because the device can be calibrated for different concentrations of iodide and the results could be stored in a flash memory for the fast detection or transmitted in cloud via low-power wide-area network (LPWAN) or 5G technologies. This, in conjunction with the low power consumption characteristics of the device, makes it viable its provisioning in Internet of Things (IoT) systems opening to both continuous monitoring and big data studies of basins and batches of seafood. Another advantage given by an input fixed voltage of low intensity consists in the possibility to generate the input signal from external sources, for example using a piezoelectric transducer, as energy harvesting approach also to supply the overall circuit. This arrangement of the circuit is possible because the voltage and current values are not high.

Considering the results obtained in iodide calibration, the voltammetric liquid sensor designed and tested, when compared to existing methods [16,17,18,19], is simpler, portable and cost effective, even if more experiments have to be performed to assess its reliability with seawater real samples. Besides, BIONOTE-L ability in discriminating the concentration of different compounds in a complex mixture, demonstrated in a test with drinkable water [30], accounts for its relevance also in seawater. 

It is worth pointing out also the ability of the BIONOTE-V to discriminate fish species. The use of a gas sensor array based on QMBs, optical techniques and many other instrumental features for fish quality and freshness assessment is of course not new [20,21,22,23,24]. Besides, this ability makes the use of BIONOTE-V reliable and relevant in the measurement process foreseen in Figure 1. Also, the possibility of developing portable sensors for seafood quality has been recently demonstrated in literature [20,21]. All the reported applications [20,21,22,23,24] are addressed to seafood quality assessment in terms of spoilage and adulteration, while this work is focus on iodide content and oriented to dietary monitoring with clinical or more generally health-control purposes.

Of course, many details have to be investigated in more depth to finalize the experiment in a functioning prototype, but this work has set up all the elements necessary to achieve the goal declared in the title.

## Figures and Tables

**Figure 1 sensors-21-04464-f001:**
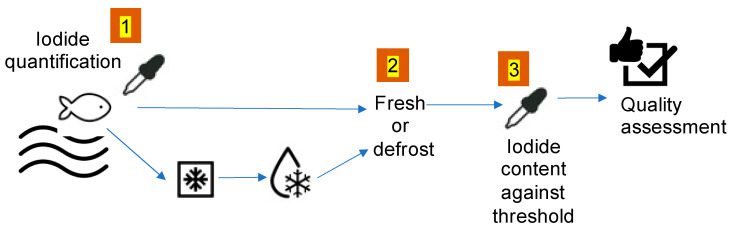
Schematic overview of a typical scenario of application of the method here tested. The multisensory instrument should be used for the evaluation of iodide content in seafood (only when fresh, meaning not defrosted); iodide content could be compared against a selected threshold; a report claiming quality assessment of seafood with respect to iodide content could be generated at the end of the process. The step labelled with numbers 1, 2 and 3 are the ones performed in this work.

**Figure 2 sensors-21-04464-f002:**
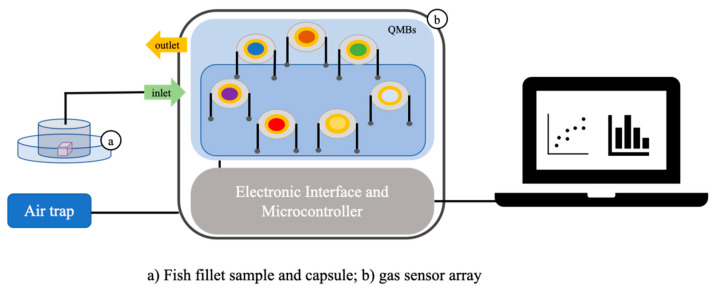
Experimental set-up of the fish headspace measurement via BIONOTE-V.

**Figure 3 sensors-21-04464-f003:**
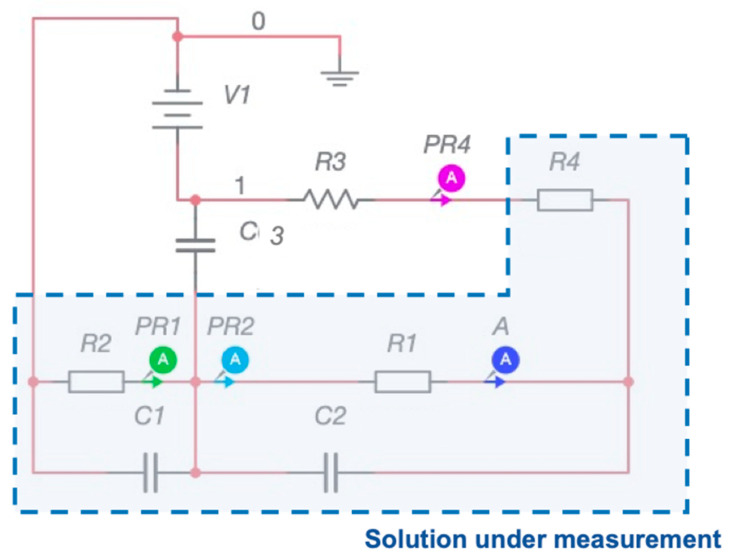
Schematic of the circuit used to control the BIONOTE-L device in a threshold modality.

**Figure 4 sensors-21-04464-f004:**
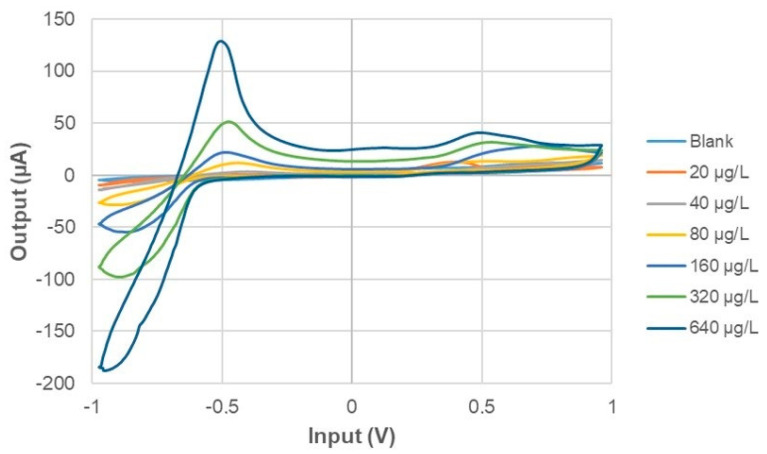
Voltammograms obtained as output current of the BIONOTE-L when calibrated with different concentration levels of iodide in the range 0–640 μg/L.

**Figure 5 sensors-21-04464-f005:**
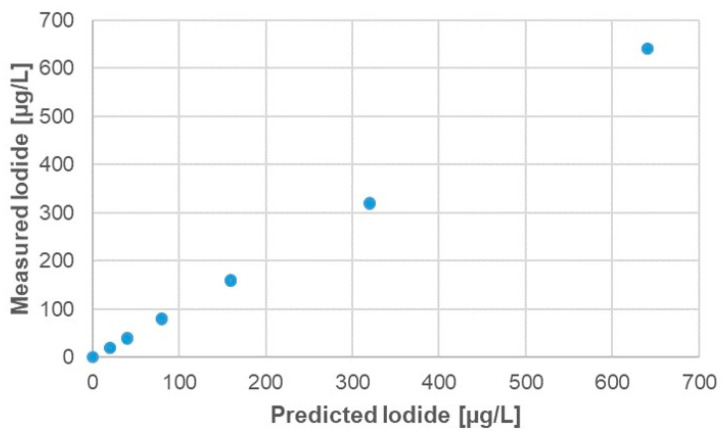
Measured vs. predicted (partial least square (PLS)-model) values of Iodide concentration. The root mean square error in cross validation (RMSECV) is 1.66 μg/L.

**Figure 6 sensors-21-04464-f006:**
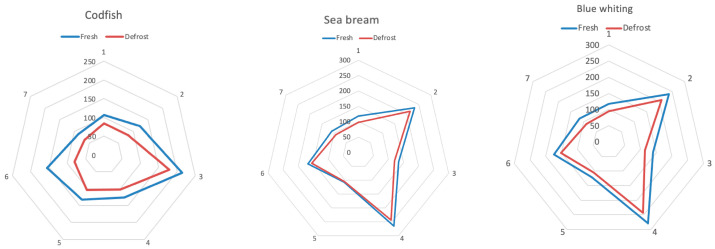
Fingerprints registered by the seven sensors of the BIONOTE-V; from left to right: codfish, sea bream, blue whiting. For each species two fingerprints are reported: (blue) fresh sample, (red) defrosted sample. Each of the 7 axes is relative to the response given by each of the QBM sensors. The response is in Hz, because it is a frequency shift with respect to the resonance frequency.

**Figure 7 sensors-21-04464-f007:**
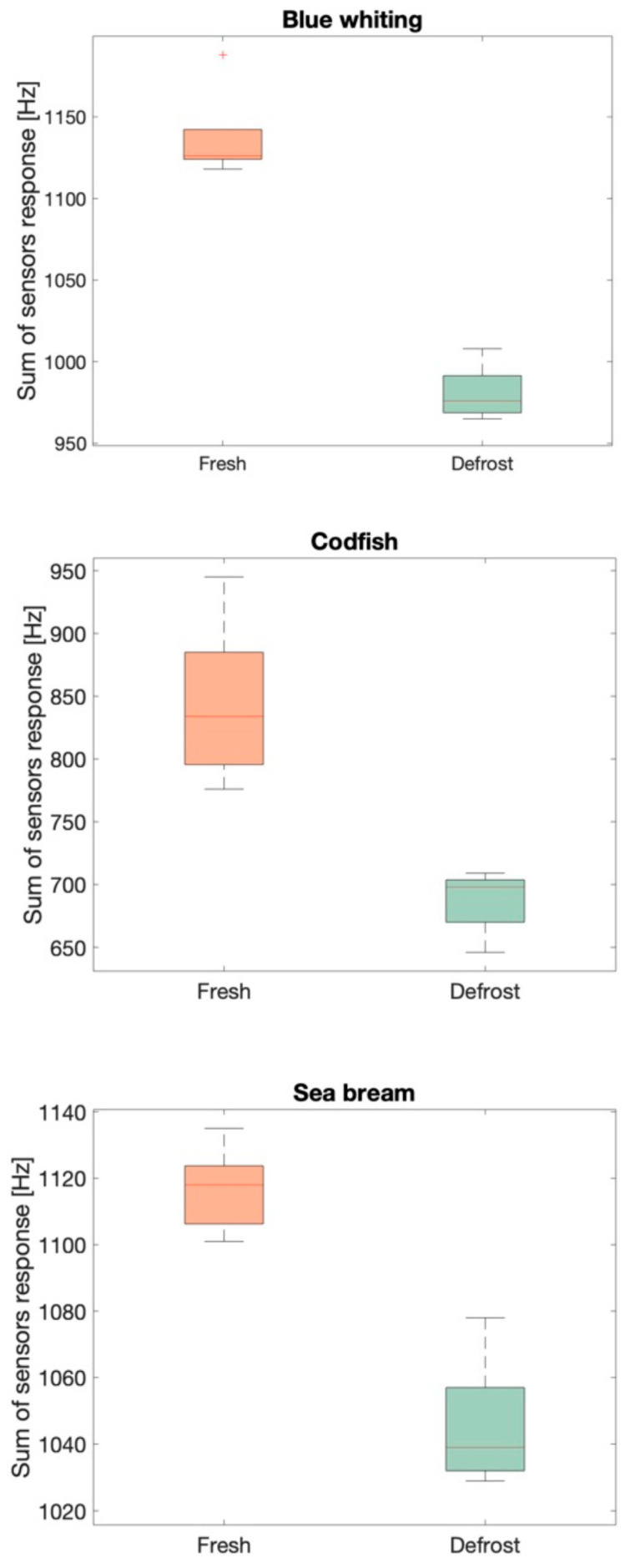
Boxplots of the olfactive intensity for each fish species (from up to down) blue whiting, codfish, sea bream, divided into two populations: fresh and defrosted. The boxes (green for defrost and orange for fresh) represents the 25–75% quartiles of each population. The median is shown with a horizontal line inside the box. The minimal and maximal values are shown with short horizontal lines (“whiskers”).

**Figure 8 sensors-21-04464-f008:**
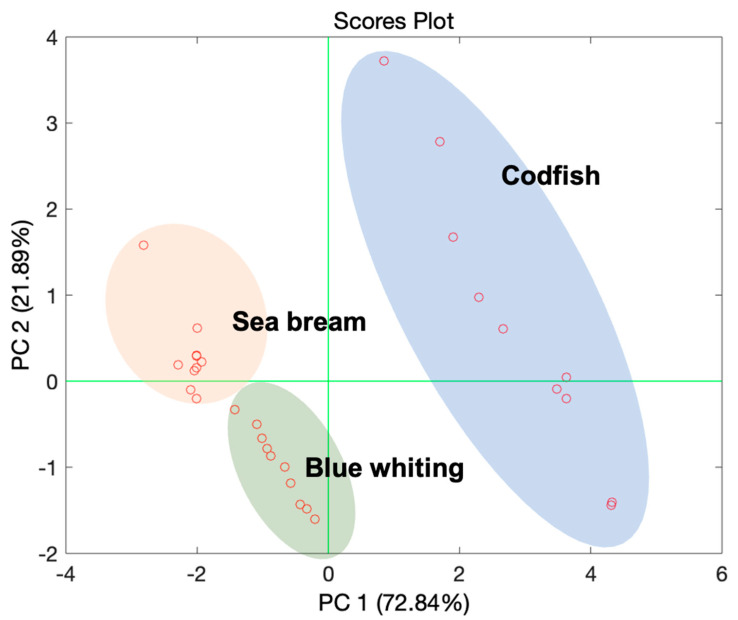
Scores plot of the first two principal components (PCs) of the PCA model elaborated on the dataset registered by the BIONOTE-V in the analysis of the samples of fishes pertaining the three species considered for the experiment: codfish, sea bream, blue whiting. For each of them a specific cluster has been defined on the PCs plane.

**Figure 9 sensors-21-04464-f009:**
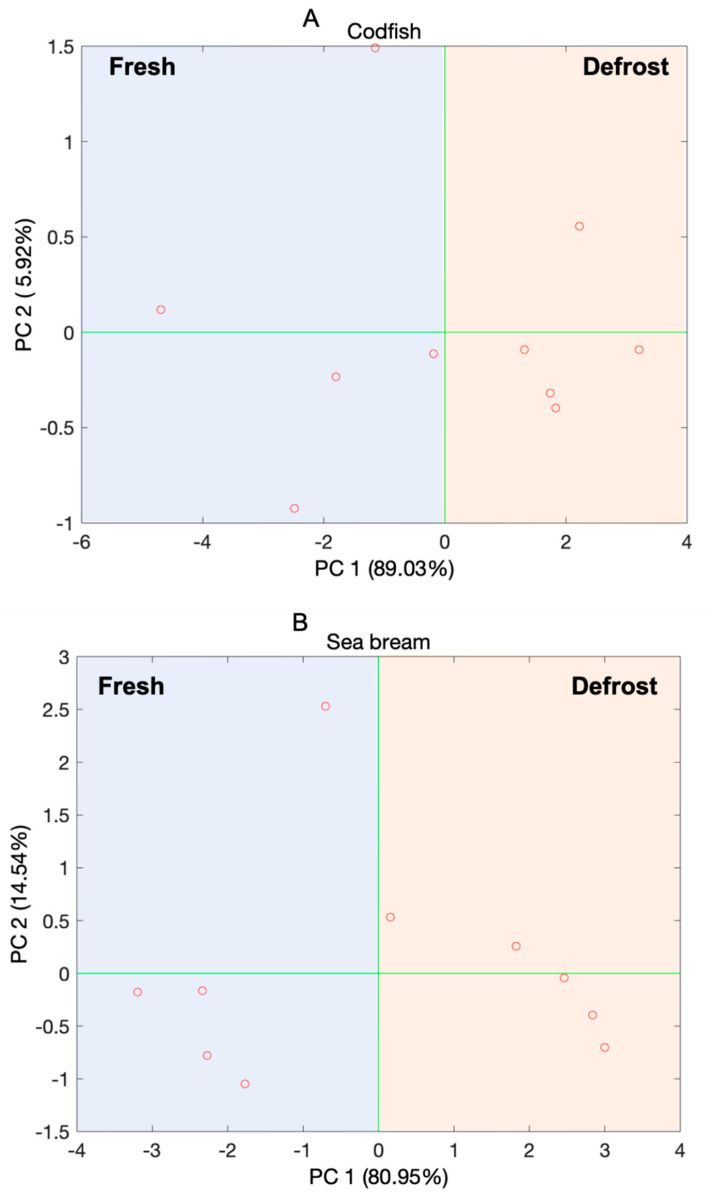
Scores plot of the first two principal components (PCs) of the PCA models elaborated on the dataset registered by the BIONOTE-V in the analysis of the samples of fishes pertaining the three species considered for the experiment: (**A**) codfish; (**B**) sea bream; (**C**) blue whiting. Along the most informative PC (the PC1) a clear discrimination can be observed among the fresh and defrosted samples of each species.

**Table 1 sensors-21-04464-t001:** Simulation results for applied voltage of −0.87 V assuming R_2_ = 305 Ω, R_4_ = 6.50 kΩ.

Applied Voltage (V) ^1^	R_1_ (Ω)	I_R1_ (μA)
−0.87	300 K	−2.27
	170 K	−4.95
	80 K	−10.3
	45 K	−16.92
	20 K	−32
	8 K	−58
	200	−120

^1^ R_2_ = 305; R_4_ = 6.50 K.

**Table 2 sensors-21-04464-t002:** Simulation results for applied voltage of −0.5 V assuming R_2_ = 305 Ω, R_4_ = 3.2 kΩ.

Applied Voltage (V) ^1^	R_1_ (Ω)	I_R1_ (μA)
−0.5	2.7 M	0.19
	232 K	2.11
	46 K	10.3
	19 K	22.15
	6.5 K	48.59
	200	128.43

^1^ R_2_ = 305; R_4_ = 3.2 K.

## Data Availability

The data presented in this study are available on request from the corresponding author.

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
