# Peer review of "A Multi-Sensor System for Sea Water Iodide Monitoring and Seafood Quality Assurance: Proof-of-Concept Study"

_sensors, 2021, doi:10.3390/s21134464_

Round 1

Reviewer 1 Report

The work undoubtedly has application advantages. It is difficult to find a scientific novelty (unless the authors describe it better in the revised manuscript). It is also difficult to refer to the obtained results because the research was carried out on a small number of samples. I think the work should undergo a thorough improvement. Below are my other comments:

line 55-60, The authors wrote that there are many analytical methods for the determination of iodine, but did not mention what these methods are. I propose to insert a table and describe the advantages and disadvantages of these methods. This will make it easier for the reader to follow this work.

line 60-76, the authors did not emphasize what is the scientific novelty of this work. Secondly, this manuscript does not refer to whether such studies have already been carried out by other research teams.

lines 130-134, What is the synthesis of a fragrance? Please describe it better. In the case of a fragrance mixture, the intensity of the fragrance depends on many factors, including the effects of the interaction of the ingredients in the mixture. It was described quite well in the works:

a). Determination of Odour Interactions in Gaseous Mixtures Using Electronic Nose Methods with Artificial Neural Networks. Sensors 2018 18(2), 519.

b). The regular interaction pattern among odorants of the same type and its application in odor intensity assessment. Sensors 2017, 17, 1624.

c). Measurement of odor intensity by an electronic nose. J. Air Waste Manag. Assoc. 2000, 50, 1750–1758.

d). Determination of odour interactions of three-component gas mixtures using an electronic nose. Sensors 2017, 17, 2380.

line 149-150, figure 1, what do the symbols PR1, PR2, PR4 and A in this figure mean?

line 178-180, at a potential of 0.5V also a peak is observed, how can this be interpreted?

line 190-193, what was the LOD of iodide ?, What was the LOQ of iodide? What was the linear range for the determination of iodides?

lines 234-236, where does this pattern come from? Please show how it was received.

line 285-287, I miss a comparison with the results obtained by other researchers. The functionality of the proposed research method could then be assessed.

Author Response

The work undoubtedly has application advantages.

Thanks to the reviewer for her/his  valuable appreciation.

It is difficult to find a scientific novelty (unless the authors describe it better in the revised manuscript).

The novel aspects have been better evidenced in the revised version.

It is also difficult to refer to the obtained results because the research was carried out on a small number of samples. I think the work should undergo a thorough improvement. Below are my other comments.

Authors agree that in the methods section the number of measurements performed and the standard deviation have not been specified. These data have been now added in the revised text. Eight concentration values of iodide have been tested: 0, 20, 40, 80, 160, 320, 640 g/L. Three measurements have been performed for each concentration level.

line 55-60, The authors wrote that there are many analytical methods for the determination of iodine, but did not mention what these methods are. I propose to insert a table and describe the advantages and disadvantages of these methods. This will make it easier for the reader to follow this work. line 60-76, the authors did not emphasize what is the scientific novelty of this work. Secondly, this manuscript does not refer to whether such studies have already been carried out by other research teams.

We agree with the reviewer. Her/his suggestion allow authors to better place the paper into the state of the art. In the revised manuscript introduction and discussion have been enriched.

We hope reviewer does not mind if we decided not to use a table: we followed also the indications of the other reviewers (4 reviewers have commented this manuscript) and the number of figures and tables is now of 11.

lines 130-134, What is the synthesis of a fragrance? Please describe it better. In the case of a fragrance mixture, the intensity of the fragrance depends on many factors, including the effects of the interaction of the ingredients in the mixture. It was described quite well in the works:

a). Determination of Odour Interactions in Gaseous Mixtures Using Electronic Nose Methods with Artificial Neural Networks. Sensors 2018 18(2), 519.

b). The regular interaction pattern among odorants of the same type and its application in odor intensity assessment. Sensors 2017, 17, 1624.

c). Measurement of odor intensity by an electronic nose. J. Air Waste Manag. Assoc. 2000, 50, 1750–1758.

d). Determination of odour interactions of three-component gas mixtures using an electronic nose. Sensors 2017, 17, 2380.

Authors have added these valuable references in the revised text. Authors have also added a deeper specification of the term ‘olfactive intensity’ in this experiment. The approach for the analysis of fish volatiles is the multidimensional measurement based on an array of sensors: the BIONOTE-V, composed of seven QMBs, functionalized with non-selective sensing materials. This approach is typical of the electronic nose paradigm. On this basis odour is a multidimensional information which is well represented by a fingerprint. Considering that the main objective was the discrimination among two conditions (fresh/defrosted) the multidimensionality of the output data has been abruptly reduced to one dimension in order to perform the most simple statistical test to achieve the discrimination goal (simpler is the data analysis technique, more reliable is the result). This index has been defined as olfactive intensity, being the sum of the response given by all the VOCS present on the headspace of the samples analyzed. Besides, the multidimensional fingerprint is the best way to represent an odour-like data, thus, as suggested by reviewer 2, also Principal Component Analysis and Partial Least Square Discriminant Analysis have been performed.

line 149-150, figure 1, what do the symbols PR1, PR2, PR4 and A in this figure mean?

As specified in the caption: The step labelled with number 1, 2 and 3 are the ones performed in this work.

line 178-180, at a potential of 0.5V also a peak is observed, how can this be interpreted?

We agree with the reviewer. Authors decided not to include this voltage value because the ‘stability’ of the abscissa relative to the peak is not the same of the other two at -0.87V and -0.5V.

line 190-193, what was the LOD of iodide ?, What was the LOQ of iodide? What was the linear range for the determination of iodides?

Being the measurement output complex curves composed of 500 points, multivariate approach has been used: the model calculated for iodine gave a root mean square error in cross validation, which gives an indication of a resolution-like parameter. Besides, no calibration curve, allowing linearity, LOD and LOQ evaluation, has been built.

lines 234-236, where does this pattern come from? Please show how it was received.

The liquid sensor is based on cyclic voltammetry. The output response is a voltammogram, as the ones reported in the patterns reported in figure 4. The method described in ref.15 is the design and development of the sensor, including the electronic circuit which, as described in ref.15, provides the sensor with a suitable stability allowing high reproducible measurements in which, each of the 500 points of the voltammogram is significant, thus opening the way to a multivariate observation of the fingerprint. The fingerprint in this way contains information relative to different compounds. This conceptual foundation of the paper has been now better explained and other references have been added to support it.

line 285-287, I miss a comparison with the results obtained by other researchers. The functionality of the proposed research method could then be assessed.

Authors agree. These aspects have been added in the discussion section.

Reviewer 2 Report

The title of manuscript is “A multi-sensor system for sea water iodide monitoring and seafood quality assurance”. The objective of the study is not obvious.

General:

Manuscripts submitted to Sensors should neither be published previously nor be under consideration for publication in another journal. The main article types are as follows:

               Articles: The journal considers all original research manuscripts, provided that the work reports scientifically sound experiments and provides a substantial amount of new information. Authors should not unnecessarily divide their work into several related manuscripts, although short Communications of preliminary but significant results will be considered. The quality and impact of the study will be considered during peer review. The recommended length of an Article is more than 16 journal pages. The article submitted for review has 11 pages, so it does not meet the above condition.

               The Introduction of the study is too brief and provides with some general information about the aroma techniques analysis. On the other hand for several decades, studies on application of different type of techniques for detection of odor have been conducted. I suggest supplementing the Chapter with additional information related to other new methods and devices in research of VOCs detections. Additional information contained in the Introduction chapter will make the aim of the study will clearly stated. The Materials and Methods section provides the reader with enough information to repeat the experiments conducted. In the Results and discussion chapter contains information should be supplemented on discussing with the other items from the last years of publication including problems of reactions on the chemical compounds. Only the basic statistical analysis was used to describe the differences. Have the Authors attempted to use other more comprehensive statistical analyzes, e.g. principal components analysis of PCA? With such a large number of parameters tested, which may affect the characteristics examined, the Principal Component Analysis (PCA) should be used to results analyzed. More advanced statistical analysis should be performed. The use of advanced statistical methods to fully describe the relationship between the parameters studied and the aspects of the research work carried out in the presented manuscript. You can determine the strength of the influence of a particular parameter on the variance of the system. At the same time, correlation relationships between the determined parameters can be received.

The conclusions are well and were supported by the data. The literature used is appropriate but should be supplementing about the items from the last years of publication. 

Author Response

The title of manuscript is “A multi-sensor system for sea water iodide monitoring and seafood quality assurance”. The objective of the study is not obvious.

Thanks to the reviewer for her/his suggestion: the title has been changed.

General:

Manuscripts submitted to Sensors should neither be published previously nor be under consideration for publication in another journal. The main article types are as follows:

               Articles: The journal considers all original research manuscripts, provided that the work reports scientifically sound experiments and provides a substantial amount of new information. Authors should not unnecessarily divide their work into several related manuscripts, although short Communications of preliminary but significant results will be considered. The quality and impact of the study will be considered during peer review. The recommended length of an Article is more than 16 journal pages. The article submitted for review has 11 pages, so it does not meet the above condition.

Thanks to the reviewer. After the revision completed addressing all the comments, the manuscript is now of the correct length.

               The Introduction of the study is too brief and provides with some general information about the aroma techniques analysis. On the other hand for several decades, studies on application of different type of techniques for detection of odor have been conducted. I suggest supplementing the Chapter with additional information related to other new methods and devices in research of VOCs detections. Additional information contained in the Introduction chapter will make the aim of the study will clearly stated.

We agree with the reviewer. Her/his suggestion allow authors to better place the paper into the state of the art. In the revised manuscript introduction and discussion have been enriched.

The Materials and Methods section provides the reader with enough information to repeat the experiments conducted.

Thanks. Considering the multi-sensor approach and the complexity of the different methods applied authors are glad to know that the methods section is effective in reviewer’s opinion.

In the Results and discussion chapter contains information should be supplemented on discussing with the other items from the last years of publication including problems of reactions on the chemical compounds.

As stated above, discussion has been enriched, also as consequence of introduction improvement.

Only the basic statistical analysis was used to describe the differences. Have the Authors attempted to use other more comprehensive statistical analyzes, e.g. principal components analysis of PCA? With such a large number of parameters tested, which may affect the characteristics examined, the Principal Component Analysis (PCA) should be used to results analyzed. More advanced statistical analysis should be performed. The use of advanced statistical methods to fully describe the relationship between the parameters studied and the aspects of the research work carried out in the presented manuscript. You can determine the strength of the influence of a particular parameter on the variance of the system. At the same time, correlation relationships between the determined parameters can be received.

The approach for the analysis of fish volatiles is the multidimensional measurement based on an array of sensors: the BIONOTE-V, composed of seven QMBs, functionalized with non-selective sensing materials. This approach is typical of the electronic nose paradigm. On this basis odour is a multidimensional information which is well represented by a fingerprint. Considering that the main objective was the discrimination among two conditions (fresh/defrosted) the multidimensionality of the output data has been abruptly reduced to one dimension in order to perform the most simple statistical test to achieve the discrimination goal (simpler is the data analysis technique, more reliable is the result). This index has been defined as olfactive intensity, being the sum of the response given by all the VOCS present on the headspace of the samples analyzed. Besides, the multidimensional fingerprint is the best way to represent an odour-like data, thus, as suggested by reviewer 2, also Principal Component Analysis and Partial Least Square Discriminant Analysis have been performed.

The conclusions are well and were supported by the data. The literature used is appropriate but should be supplementing about the items from the last years of publication. 

Thanks to the reviewers. As stated above, this comment has been addressed.

Reviewer 3 Report

A multi-sensor system for sea water iodide monitoring and seafood quality assurance

Alessandro Zompanti, Simone Grasso, Anna Sabatini, Luca Vollero, Giorgio Pennazza, Marco Santonico

Herewith I am submitting my reviewer comments for the above-mentioned manuscript, which is under consideration to be published in Sensors. The article is about detecting and quantifying iodine in water and seafood. The purpose is to identify whether seafood is fresh or defrosted. The detection is done with QCM gas sensors.

Overall, the article is well written and clear and the language is of sufficient quality. The reported techniques are not very novel (QCM is a routine technique and ref 15 reports the same kind of functionalization) but nevertheless the information might be of value for people who monitor seafood quality.

Line 19: “which is contemporary able to evaluate iodide content in seawater” I do not know if the contemporary is grammatically correct here

Line 65: “represents an optimal dietary option for iodide assumption” should it be consumption?

Line 67: “which is contemporary able to” also here the contemporary can be removed

Line 117: “Three species of fishes” should be “Three species of fish”

I am missing information on the sensors in this paper. There is almost nothing about the QCM and just a reference for the modification. I also would like to see more reasoning why the sensor is sensitive to iodine and not some other compounds. Also what is on each QCM?

In the introduction I am missing what is really new here. There is at least ref 15 which is very close and potentially others. The literature coverage is very short given that there is already A LOT done in sensing. Both in the area of QCMs and likely also iodine sensing. I am also missing a discussion on how the sensor system is selective for iodine (and potentially how this compares to other techniques which would allow sensitivity [for example attaching MIPs, antibodies, enzymes…])

Author Response

Herewith I am submitting my reviewer comments for the above-mentioned manuscript, which is under consideration to be published in Sensors. The article is about detecting and quantifying iodine in water and seafood. The purpose is to identify whether seafood is fresh or defrosted. The detection is done with QCM gas sensors.

Overall, the article is well written and clear and the language is of sufficient quality. The reported techniques are not very novel (QCM is a routine technique and ref 15 reports the same kind of functionalization) but nevertheless the information might be of value for people who monitor seafood quality.

Thanks to the reviewer for her/his valuable appreciation of the clarity and relevance of the work. Regarding novelty, authors agree that QCM is not an innovative working principle. The novelty here proposed is the utilization of gas sensors (based on QCM) and liquid sensors (based on cyclic voltammetry) for the evaluation of different features regarding the sample and its environment. The authors, anyway, taking in consideration reviewer’s comment on novelty, have better specified this aspect in the introduction.

Line 19: “which is contemporary able to evaluate iodide content in seawater” I do not know if the contemporary is grammatically correct here

Probably is not consistent: deleted.

Line 65: “represents an optimal dietary option for iodide assumption” should it be consumption?

Corrected, thanks.

Line 67: “which is contemporary able to” also here the contemporary can be removed

As above, authors agree, thanks.

Line 117: “Three species of fishes” should be “Three species of fish”

Corrected, thanks.

I am missing information on the sensors in this paper. There is almost nothing about the QCM and just a reference for the modification. I also would like to see more reasoning why the sensor is sensitive to iodine and not some other compounds. Also what is on each QCM?

More info on QCM sensors have been added in the methods. The sensing material covering the QCM are given in the cited reference, but authors agree it is useful to repeat these data in this manuscript. QCM sensors are used for volatiles analysis addressed to fresh/defrosted discrimination, not for iodine; probably it is not clear enough and authors have better clarified this point in the revised text.

In the introduction I am missing what is really new here. There is at least ref 15 which is very close and potentially others. The literature coverage is very short given that there is already A LOT done in sensing. Both in the area of QCMs and likely also iodine sensing. I am also missing a discussion on how the sensor system is selective for iodine (and potentially how this compares to other techniques which would allow sensitivity [for example attaching MIPs, antibodies, enzymes…])

We agree with the reviewer. Her/his suggestion allow authors to better place the paper into the state of the art. In the revised manuscript introduction and discussion have been enriched.

Reviewer 4 Report

As correctly noted by the authors, Iodine is a trace chemical element fundamental for a healthy human organism. Iodine is present in seawater and seafood, and its common forms in the diet are iodide and iodate. A multisensor instrument which is contemporary able to evaluate iodide content in seawater and seafood and to discriminate when the seafood is fresh or defrosted quality, is strategical for seafood quality assurance. Therefore, the topic of this work is undoubtedly relevant.

Nevertheless, I cannot recommend the work for publication in its current form for the following reasons:

  1. The authors demonstrated the possibility of electrochemical determination of iodide on a series of solutions prepared in deionized water, and developed a prototype of an express tester. However, in the future, they propose to use the developed prototype for sea water iodide monitoring and seafood quality assurance (as stated in the title of the article). A large number of other substances, including chlorides, are present in seawater and in real samples. Moreover, the content of chlorides is several orders of magnitude higher than the content of bromides. It would be extremely important to first assess at least the interfering effect of other halides on the analytical signal of the system. Only then can a prototype be constructed for simplified measurements. It is possible that it is the change in chloride concentration after defrosting that affects the observed results in Fig. 4 (line 250) and Fig. 5 (line 259), because the signal after defrosting is always lower.
  2. It is not clear from the text how the "fingerprints" of the samples obtained according to the previously described method (ref. 15) correlate with the results of the determination of bromide on the created prototype of the electrochemical sensor. Why should they correlate?
  3. There is confusion in the numbering of figures in the manuscript. Numbers 4 and 5 are duplicated, denoting completely different data, and this makes the perception of the text extremely difficult.
  4. The trend line and the error range should be indicated in addition to the points in Fig. 5 (line 189). It is much clearer than just a number in the text.
  5. Probably on line 208 there is a typo in the word "relative".

Author Response

As correctly noted by the authors, Iodine is a trace chemical element fundamental for a healthy human organism. Iodine is present in seawater and seafood, and its common forms in the diet are iodide and iodate. A multisensor instrument which is contemporary able to evaluate iodide content in seawater and seafood and to discriminate when the seafood is fresh or defrosted quality, is strategical for seafood quality assurance. Therefore, the topic of this work is undoubtedly relevant.

Authors thank the reviewer for her/his valuable appreciation

Nevertheless, I cannot recommend the work for publication in its current form for the following reasons:

Authors have taken into consideration all the reasons reported below, consequently modifying the manuscript addressing reviewer’s comments.

  1. The authors demonstrated the possibility of electrochemical determination of iodide on a series of solutions prepared in deionized water, and developed a prototype of an express tester. However, in the future, they propose to use the developed prototype for sea water iodide monitoring and seafood quality assurance (as stated in the title of the article). A large number of other substances, including chlorides, are present in seawater and in real samples. Moreover, the content of chlorides is several orders of magnitude higher than the content of bromides. It would be extremely important to first assess at least the interfering effect of other halides on the analytical signal of the system. Only then can a prototype be constructed for simplified measurements. It is possible that it is the change in chloride concentration after defrosting that affects the observed results in Fig. 4 (line 250) and Fig. 5 (line 259), because the signal after defrosting is always lower.

We agree with reviewer’s observation: in this paper a proof of concept is presented, thus, in this paper, only iodine identification has been demonstrated with specific calibration tests. Besides, the BIONOTE-L has been already used for water characterization (Santonico, M., Parente, F. R., Grasso, S., Zompanti, A., Ferri, G., D'Amico, A., & Pennazza, G. (2016). Investigating a single sensor ability in the characterisation of drinkable water: a pilot study. Water and Environment Journal, 30(3-4), 253-260). In this paper the single sensor has shawn to be able to discriminate the concentration of numerous salts present in the same water sample. Considering the intensity of the output responses for those salts and for iodine, authors have decided to present here iodine calibration and, thanks to this previous experiment, to postpone iodine calibration in complex mixtures (mimicking seawater composition) to  a future work.

We do not know how defrosting influences iodine detection, because we decided to use the fresh/defrosted discrimination as a ‘traffic-light’: only fresh samples have to be evaluated in terms of iodine content.

  1. It is not clear from the text how the "fingerprints" of the samples obtained according to the previously described method (ref. 15) correlate with the results of the determination of bromide on the created prototype of the electrochemical sensor. Why should they correlate?

As stated above authors have previously tested this sensor with many different liquids, including water. The method described in ref.15 is the design and development of the sensor, including the electronic circuit which, as described in ref.15, provides the sensor with a suitable stability allowing high reproducible measurements in which, each of the 500 points of the voltammogram is significant, thus opening the way to a multivariate observation of the fingerprint. The fingerprint in this way contain information relative to different compounds. This conceptual foundation of the paper has been now better explained and other references have been added to support it.

  1. There is confusion in the numbering of figures in the manuscript. Numbers 4 and 5 are duplicated, denoting completely different data, and this makes the perception of the text extremely difficult.

Authors are very sorry for this inconvenient. Figures numbering has been corrected.

  1. The trend line and the error range should be indicated in addition to the points in Fig. 5 (line 189). It is much clearer than just a number in the text.

Figure 5 is the plot of the real values of concentration vs the ones predicted by the PLS model. It is not a fitting. The error in calibration is the one given by the model: the RMSEVC, which is of 1.66 ug/L. If the reviewer refers to the error bar in the measurements of the different concentrations, authors agree that in the methods section the number of measurements performed and the standard deviation have not been specified. These data have been now added in the revised text.

  1. Probably on line 208 there is a typo in the word "relative".

Thanks, corrected.

Round 2

Reviewer 1 Report

I am satisfied with the answers given by the authors. However, I believe that on the basis of Figure 4 it is possible to determine the LOD for the iodide content. The authors only gave the measuring range from 0 to 640. Moreover, I asked the authors to present where the formula in lines 271-273 comes from. How was the system sensitivity determined?

Author Response

I am satisfied with the answers given by the authors. However, I believe that on the basis of Figure 4 it is possible to determine the LOD for the iodide content. The authors only gave the measuring range from 0 to 640. Moreover, I asked the authors to present where the formula in lines 271-273 comes from. How was the system sensitivity determined?

Thanks to the reviewer for her/his valuable appreciation and suggestions.

Authors agree that the LOD can be estimated, but only theoretically. Considering that The RMSECV has been estimated by a multidimensional approach, the LOD has been calculated, as suggested by the reviewer, looking at figure 4, focusing on the current responses obtained for the peaks of interest. At this point, authors realized that this suggestion of the reviewer could be very useful to understand the performance of the threshold-device strategy, which exploits, indeed, the mono-dimensional current responses for each voltage input. Looking at table 1 and 2 the concentration-current curve is linear and the sensitivity is of 0.18 ug/L. Considering the minimum detectable current as 0.1 uA (calculated considering the Johnson Thermal Noise), the resolution is of 0.55 ug/l, which gives a final LOD of 1.66 ug/L, the same value of the estimated RMSECV in the multidimensional approach.

The formula in lines 271-273 is the derivate of the output voltage of the circuit calculated with respect to the variable R.

Reviewer 2 Report

The authors referred to the comments from the previous review for the manuscript titled: A multi-sensor system for sea water iodide monitoring and seafood quality assurance. The introduction part still needs to be completed. I suggest supplementing the Chapter with additional information related to other new methods and devices in research of VOCs detections. I accept explanations in the later part of the manuscript. In the future, I suggest using more precise describing relationships between the parameters studied. They supplemented the discussion with a new literature data strengthens the message and importance of information in the manuscript. 

Author Response

The authors referred to the comments from the previous review for the manuscript titled: A multi-sensor system for sea water iodide monitoring and seafood quality assurance. The introduction part still needs to be completed. I suggest supplementing the Chapter with additional information related to other new methods and devices in research of VOCs detections. I accept explanations in the later part of the manuscript. In the future, I suggest using more precise describing relationships between the parameters studied. They supplemented the discussion with a new literature data strengthens the message and importance of information in the manuscript. 

Thanks to the reviewer for her/his valuable comments. The introduction has been improved as suggested.

Reviewer 3 Report

The authors have significantly improved their paper and included the comments. I think it is good now. 

Author Response

The authors have significantly improved their paper and included the comments. I think it is good now. 

Thanks to the reviewer for her/his valuable contribution to the improvement of the paper quality

Reviewer 4 Report

The authors took into account the comments made and significantly improved the manuscript. The revised title accurately reflects the content of the work done. Principal Components Analysis significantly enriched the information obtained by means of the sensor prototype. The reviewer hopes that in future work an assessment of the crosstalk effect of other ions (primarily halides) on the analytical signal will be performed.

I can recommend the work for publication after fixing minor bugs:

  1. The term "the liquid sensor" (lines 82, 374, 386) is not the best.
  2. Extra space in line 389.
  3. Reference 23 refers to a website in the context of the properties of deionized water. According to the reviewer's opinion, the authors can find a more reliable literature source with DOI.

Author Response

The authors took into account the comments made and significantly improved the manuscript. The revised title accurately reflects the content of the work done. Principal Components Analysis significantly enriched the information obtained by means of the sensor prototype. The reviewer hopes that in future work an assessment of the crosstalk effect of other ions (primarily halides) on the analytical signal will be performed.

Thanks to the reviewer for her/his valuable comments.

I can recommend the work for publication after fixing minor bugs:

  1. The term "the liquid sensor" (lines 82, 374, 386) is not the best.

Thanks to the reviewer. It has been changed in ‘voltammetric liquid sensor’

  1. Extra space in line 389.

Ok corrected

  1. Reference 23 refers to a website in the context of the properties of deionized water. According to the reviewer's opinion, the authors can find a more reliable literature source with DOI.

Ok, it has been changed.